# Effect of Land Use and Land Cover Change on Soil Erosion in Erer Sub-Basin, Northeast Wabi Shebelle Basin, Ethiopia

**Gezahegn Weldu Woldemariam** [1,*]  **and Arus Edo Harka** [2]

[1] Geoinformation Science Program, School of Geography and Environmental Studies, Haramaya University, P.O. Box 138, 3220 Dire Dawa, Ethiopia

[2] Hydraulic and Water Resources Engineering Department, School of Water Resources and Environmental Engineering, Haramaya Institute of Technology (HiT), Haramaya University, P.O. Box 138, 3220 Dire Dawa, Ethiopia; harqaa@gmail.com

* Correspondence: gezahegnw3@gmail.com; Tel.: +251-091-096-1491

**Abstract:** Land use and land cover change (LULCC) is a critical factor for enhancing the soil erosion risk and land degradation process in the Wabi Shebelle Basin. Up-to-date spatial and statistical data on basin-wide erosion rates can provide an important basis for planning and conservation of soil and water ecosystems. The objectives of this study were to examine the magnitude of LULCC and consequent changes in the spatial extent of soil erosion risk, and identify priority areas for Soil and Water Conservation (SWC) in the Erer Sub-Basin, Wabi Shebelle Basin, Ethiopia. The soil loss rates were estimated using an empirical prediction model of the Revised Universal Soil Loss Equation (RUSLE) outlined in the ArcGIS environment. The estimated total annual actual soil loss at the sub-basin level was 1.01 million tons in 2000 and 1.52 million tons in 2018 with a mean erosion rate of 75.85 t ha$^{-1}$ y$^{-1}$ and 107.07 t ha$^{-1}$ y$^{-1}$, respectively. The most extensive soil loss rates were estimated in croplands and bare land cover, with a mean soil loss rate of 37.60 t ha$^{-1}$ y$^{-1}$ and 15.78 t ha$^{-1}$ y$^{-1}$, respectively. The soil erosion risk has increased by 18.28% of the total area, and decreased by 15.93%, showing that the overall soil erosion situation is worsening in the study area. We determined SWC priority areas using a Multi Criteria Decision Rule (MCDR) approach, indicating that the top three levels identified for intense SWC account for about 2.50%, 2.38%, and 2.14%, respectively. These priority levels are typically situated along the steep slopes in Babile, Fedis, Fik, Gursum, Gola Oda, Haramaya, Jarso, and Kombolcha districts that need emergency SWC measures.

**Keywords:** LULCC; SWC; soil erosion risk; Erer Sub-Basin; RUSLE; ArcGIS; SWC; MCDR

## 1. Introduction

Soil erosion is a complex three-phase dynamic process involving detachment and transport of the particles or aggregate topsoil by the physical forces of wind, water, and gravity (mass movement) and immediate sediment deposition in downstream areas [1–8]. Water-induced soil erosion is indeed the most important land degradation problem worldwide [3–5]. Soil erosion has been documented as one of the greatest global problems that result in serious threats to natural resources, agriculture, and the environment [1–7]. Erosion displaces soil organic carbon and the most important nutrients, and consequently affects vegetation growth, biodiversity, and overall sustainability of ecosystem services and functions [2–9]. Soil erosion can also cause severe environmental problems, including soil and water degradation, a decrease in land productivity, and eutrophication and sedimentation of water bodies [3–11]. Numerous studies have reported that the magnitude of soil erosion rates has been accelerating worldwide due to land use and land cover change (LULCC) and inappropriate land use

and management practices resulting in widespread land degradation process [2–5,12–18]. The global annual average potential soil loss due to water-caused erosion was estimated at 35 billion tons in 2001 [4]. LULCCs have been accounted for an overall increase of 2.5% in the global average soil erosion between 2001 and 2012 [4]. According to the study by the Global Soil Partnership (GSP) [13], around 75 billion tons of topsoil is lost annually due to erosion from the arable land worldwide that is equivalent to about $400 billion losses in agricultural production. In connection to this, the Food and Agriculture Organization (FAO) of the United Nations and Intergovernmental Technical Panel on Soils [14] stated that "if action is not taken to reduce erosion, total crop yield losses projected by the year 2050 would be equivalent to removing 1.5 million km$^2$ of land from crop production–or roughly all the arable land in India". In the developing countries where the overall economy and the livelihood of a majority of the population depend on the productivity of their land, the displacement of the most productive topsoil layer by erosion and a poor conservation practices have resulted in the reductions in agricultural production and land productivity potential and contributing to food insecurity [18,19].

With a population of about 107.53 million (estimated as of December 2018) growing at an annual rate of 2.46%, Ethiopia is the most populous landlocked country in the continent of Africa, and the second-most populous nation in Africa [20]. Agriculture sector, which accounts for about 50% of the Gross Domestic Product (GDP), 85% of the total export revenue, and over 80% of the total employment, is the main source of the country's economy [21–25]. The great majority of the population is dependent on subsistence agriculture that is an overwhelming vulnerable to the recurrent droughts and land degradation [21–28]. Rapid population increase and growing demand posed a greater pressure on land resources, leading to severe soil erosion and land degradation in various parts of the country. To cope with the worsening environmental problems, a series of Soil and Water Conservation (SWC) programs have been launched in Ethiopia since the 1970s and 1980s [29]. Despite conservation measures taken over the past decades, land degradation is continued to threaten crop production and land productivity potential, and negatively affecting livelihood systems, food security, and the country's economy [29–32]. It was estimated that the land degradation cost to an annual agricultural GDP range from 2% to 6.75% [21]. The loss of topsoil by water erosion in Ethiopia was estimated at 1.5 billion tons per annum with a mean erosion rate of 42 t ha$^{-1}$ y$^{-1}$ [27,28]. However, the magnitude of soil erosion rates varies across the physiographical regions in the country.

The Ethiopian highland, which covers about 44% of the country's total geographical area and sustains the livelihood of about 87% of the population, is the most eroded physiographical regions in the country [33–35]. The estimated annual soil loss from the highland areas varies widely from 200 t ha$^{-1}$ y$^{-1}$ to as high as 300 t ha$^{-1}$ y$^{-1}$ [36–40]. Intense rainfall, low vegetation cover, rugged topography, and anthropogenic factors are thought to be the most important factors contributing to a higher rate of soil erosion. Deforestation, agriculture land and urban expansion, cultivation in upslope areas, uncontrolled and overgrazing were the major anthropogenic drivers of soil erosion in the highland areas of the country [27,38,40]. A report from the Soil Conservation Research Program (SCRP) indicates that almost 50% of the Ethiopian highlands were seriously eroded, while 4% of the highland areas have reached a level of irreversibility that they will no longer give economic productivity in the foreseen future [38,39].

Assessing and mediating the untoward effects of soil erosion risk while increasing productivity of land resources has become the key concern of policymakers and conservation planners around the world [5,11,15–18,41–43]. In order to control erosion risk at river basin and watershed scales, there is a need to predict spatially distributed rates of soil erosion and sediment yield [18,44,45]. Given the complexity of interplays among and within the physical and hydrological factors that involved soil erosion (e.g., topography, rainfall, vegetation cover, soil, and land use) and soil conservation practices, consistent estimation of soil loss rates in the river basins and watersheds remains a key challenge in soil erosion study [17,46–48]. The integration of hydrological models with a comprehensive geospatial dataset on the biophysical and the hydrological driven factors that cause soil erosion has been recognized as a promising approach for estimation of soil loss and sediment yield.

Over the past decades, numerous hydrological models ranging from relatively simple empirical models to more complex physically based prediction models have been developed for the derivation of spatially variable factors and estimating their combined effect on soil erosion and sediment yield [49–63]. As compared with the physical-based models, the empirical models are the widely used prediction tools due to their minimal data required and ease of application to estimate soil loss rates at a regional and global scale [4,44]. Among these models, the Revised Universal Soil Loss Equation (RUSLE) [54], which is a derivative of the Universal Soil Loss Equation (USLE) [50], is the most frequently applied model for predicting the long-term average annual soil loss caused by raindrop splash and runoff [63,64]. Recently with the advancement of satellite remote sensing and Geographic Information Systems (GIS), the adoptability of an empirical prediction model of RUSLE is considerably enhanced and soil erosion assessment at different spatial and temporal scales has become possible [4,45]. The RUSLE model have been extensively used in the various parts of the world for soil loss estimation and conservation planning by assimilating them with remotely sensed data and GIS method [2–5,11,64,65].

In the Upper Wabi Shebelle Basin, which is located in Ethiopia, soil erosion and land degradation have become serious environmental problems over recent decades. The combination of LULCC, steep slopes, climate, and unsustainable land management practices were found to be the influential factors aggravating the erosion problem at different scales [18,66–75]. Up-to-date spatial and statistical data on basin-wide erosion rates can provide an important basis for planning and conservation of soil and water ecosystems. Studies previously conducted in the Upper Wabi Shebelle Basin typically covered small catchment or watershed, and focused the assessment of soil loss, runoff, sediment yield, and groundwater recharges [18,67–75]. Such studies have been mainly supported by remote sensing data and GIS-based hydrological models. For instance, Senti et al. [71] examined soil erosion and sediment yield in the Lake Haramaya Catchment of eastern Ethiopia by using the Soil and Water Assessment Tool (SWAT) and Modified Universal Soil Loss Equation (MUSLE) models. They found that the anthropogenic drivers were major causes attributed to severe soil erosion occurring in the catchment [71]. Moreover, Woldemariam et al. [18] applied the RUSEL, GIS, and a Multi Criteria Decision Rule (MCDR) method to identify priority areas for SWC measures based on the severity levels of soil erosion risk in the Gobele Watershed, East Hararghe Zone, Ethiopia. Selecting the Lafto watershed in the Upper Wabi Shebelle Basin as their study area, Ayala et al. [68] applied the SWAT model to investigate the sensitivity of rainfall-runoff and sediment yield to SWC measures. Likewise, Megersa [69] conducted a similar study with an emphasis on Erer-Guda catchment and reported that the magnitude of rainfall-runoff and sediment yield was considerably higher on cultivated land than in another land covers. Furthermore, Gebere et al. [75] examined the impact of LULCC on the groundwater recharges of the Lake Haramaya Watershed in the East Hararghe Ethiopian highland. However, none of the past studies addressed how the patterns and the process of LULCC have changed the spatial extent of soil erosion risk over the past decades. Therefore, this study was intended to (i) assess the magnitudes of LULCC between 2000 and 2018; (ii) examine consequent spatial changes among erosion risk categories; and (iii) identify priority areas for SWC based on the severity levels of soil loss and erosion risk in the Erer Sub-Basin, Wabi Shebelle Basin, Ethiopia. The findings of the present study can provide an important foundation in planning a future intervention to minimize the untoward impacts of soil and water resource degradation in the study area. This study is also important to land degradation neutrality voluntary national target and strategy of Ethiopia [31], which is aimed to attain the land degradation neutral environment throughout the country by the year 2040.

## 2. Materials and Methods

### 2.1. Description of the Study Area

The Wabi Shebelle Basin is one of the transboundary river basins in East Africa and a highly important basin in Ethiopia. This study was carried out in the Erer Sub-Basin within the Upper Wabi Shebelle Basin, which is located, geographically, between 08°12′35″ N to 09°31′07″ N latitude and

42°04′27″ E to 42°31′07″ E longitude with an elevation range of 800–2920 meters above mean sea level (Figure 1). The drainage area of the Erer Sub-Basin is about 3860 km² of which, about 73.5% is classified as Kolla (warm semiarid), which ranges from 500 to 1500 meters, while Woinadega (cool sub-humid; 1500–2300 meters) and Dega (cool humid; 2300–3200-meters) account for about 25.12% and 1.36%, respectively, of the total drainage area [76]. The mean annual rainfall ranges between 744 and 1017 mm (based on data from three meteorological stations: Kombolcha, Babile, and Bisidimo) and mostly occurs during summer [77]. The mean monthly maximum temperature reaches up to 29.95 °C and a mean monthly minimum air temperature reaches up to 16.72 °C [77]. The dominant soil types include Calcaric regosols, Eutric nitosols, Eutric regosols, Dystric cambisols, Haplic xerosols, and Humic cambisols, with a proportion of each class contributing 4%, 8%, 20%, 19%, 49%, and 16%, respectively, of the total study area [78].

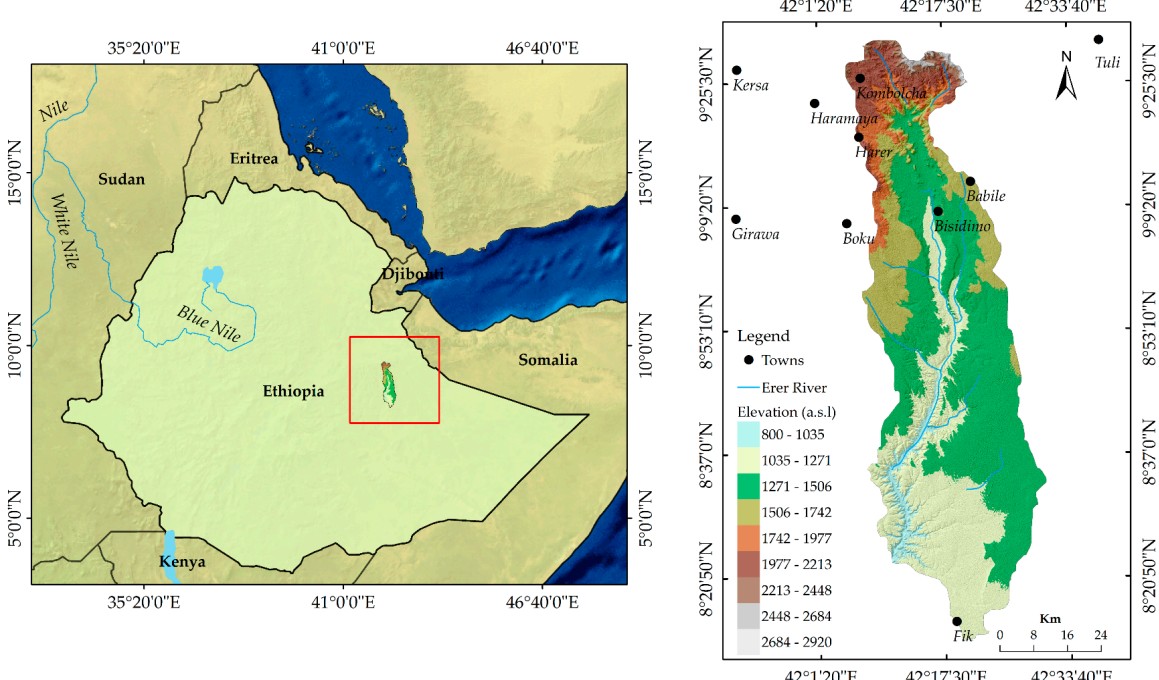

**Figure 1.** Location of the Erer Sub-Basin, North East Wabi Shebelle Basin, Ethiopia.

## 2.2. Data Collection

Four geospatial datasets collected from different sources were used in the present study, namely: rainfall data, Landsat satellite imagery, digital elevation model (DEM), and soil classification map. The average annual rainfall data for the period of twenty years (1998–2018) with fifteen meteorological stations (Babile, Bedeno, Boku, Bisidimo, Fik, Girawa, Haramaya, Harer, Jijiga, Kersa, Kombolcha, Kulubi, Legehida, Majo Weldya, and Tuli) was obtained from the National Meteorological Agency (NMA) of Ethiopia [77]. We used multispectral satellite data from Landsat 5 Thematic Mapper (TM) image (Path 166/Row 54) acquired on14 January 2000 and Landsat 8 Operational Land Imager (OLI) image (Path 166/Row 54) acquired on 20 March 2018. The Landsat images were retrieved from the United States Geological Survey (USGS) website via Landsat Look Viewer [79]. Moreover, the field survey and observations were conducted during January–March 2018 to collect ground truth data correspond to LULC classes of interest throughout the study area. We used a handheld Global Positioning System to mark the spatial locations of the reference data. Due to a constraint of field data, Google Earth Image was employed to collect reference samples for the 2000 image classification and accuracy assessment. LULC classes of the samples include bare land, cropland, forestland, settlement, shrubland, and water bodies. We identified the sampled LULC classes based on the field survey and previous experience about the study area. A total of 450-ground truth data was collected for the

two-study period from the field stratified randomly to LULC classes and the high-resolution Google Earth image.

The DEM of a 30m pixel size was provided by the Ministry of Economy, Trade, and Industry of Japan and the National Aeronautics and Space Administration (NASA) [80]. In addition, the soil classification map and the attribute value of the soil classes were downloaded from the FAO Harmonized World Soil Database (HWSD) in the Environmental System Research Institute (ESRI) shapefile format [78]. A description of the soil classes is given in Table S1.

### 2.3. Methods

### 2.3.1. Delineation of the Sub-Basin Area

We performed the raster analysis based on the terrain data of the DEM [80] with a grid resolution of 30 m × 30 m and delineated the Erer Sub-Basin boundary using the Arc-Hydro extension tools in the ArcGIS software version 10.5 (Environment Systems Research Institute (Esri), Inc. Redlands, CA, USA).

### 2.3.2. LULC Classification

The LULC data of the Erer Sub-Basin was interpreted using satellite imagery from TM and OLI sensors. The two Landsat images were preprocessed to correct the inherent geometric, radiometric, and atmospheric distortion to produce more accurate interpretation results with actual ground scenes representation [81,82]. Of the spectral bands, each single-band image in the visible (blue, green, and red) and near infrared (NIR), and shortwave infrared (SWIR) spectral bands of TM (1–5,7) and OLI (2–7) sensors, with a 30 m pixel size, were combined to develop a multi-band composite images [83]. These spectral bands were chosen for their values in discriminating soil/vegetation, water body, and urban features [83]. A portion of Landsat images covering an area of interest (AOI) was extracted using the vector shapefile of the study area and the subset tool in ERDAS IMAGINE® software version 2015 (Intergraph Corporation, Huntsville, AL, USA). A preprocessed Landsat satellite images were classified into separate maps of LULC classes using a pixel-based supervised maximum likelihood classifier (MLC) approach. Based on Level 1 of the Anderson classification system [84], the six LULC classes identified in the study area—bare land, cropland, forestland, settlement, shrubland, and water body—have been classified for the 2000 and 2018 images separately.

Prior to change detection analysis one should assess the classification accuracy of LULC data generated from remotely sensed data to check the level of agreements between the reference samples and the classified images. In this study, the accuracies of the classified LULC image for 2000 and 2018 were validated using ground truth data. Out of 450-ground truth data generated based on the stratified random sampling method for the LULC classes, 150 reference points were used as training data for image classification. The remaining 300 references were used to validate the accuracy of the classified satellite image of the respective years. Overall accuracy, user and producer accuracies, and the Kappa (Kˆ) coefficient were generated from the error matrices.

### 2.3.3. LULCC Analysis

The classified LULC imagery for 2000 and 2018 were overlaid in order to drive a cross-tabulation matrix showing the spatial conversions among LULC categories between 2000 and 2018. The diagonal entries indicate the amount of LULC categories that remained unchanged between Time 1 and Time 2, whereas the off-diagonal elements account for a conversion from one class to another LULC classes [85,86]. The change detection matrix was further analyzed in order to calculate gain, loss, persistence, net change, total change, swap, and gain to persistence, loss to persistence, the net change to persistence for each LULC category between Time 1 and Time 2 [85–87]. The loss column represents the amount of loss for a LULC category i between Time 1 and Times 2, while the gain row indicates the amount of gain for a LULC category j between the same periods [85]. The swap change incorporates the amount of both loss and gain to account for a LULC category lost in a given site to the corresponding

gained in another site [85]. The computation of the swap change for a LULC category j requires pairing a grid cell of both gain (i.e., the differences between the column totals and persistence) and loss (i.e., the differences between the row totals and persistence) of a land category j (Equation (1)) [85].

$$S_j = 2\min\left(P_{j+} - P_{jj}; P_{+j} - P_{jj}\right) \tag{1}$$

where $S_j$ is the amount of swap; $P_{j+}$ is a column sum of a land cover category; $P_{jj}$ is the amount of persistence in a land cover category; and $P_{+j}$ is the sum row amount of a land cover category.

### 2.3.4. Determination of the RUSLE Factors

The empirical prediction model of RUSLE is a widely applied tool to estimate the long-term average annual soil loss from hillslopes due to rainstorm power and runoff [63]. Over the recent decade, the RUSLE and its adapted versions have been successfully tested at various hydrological basins and watersheds under different topography, climate, soil, and land-cover conditions [4,5,16–18,64,65]. In this study, the RUSLE model is chosen due to its adaptability at different spatial scales with a relatively minimal data required and easy to integrate with the ArcGIS environment for predicting soil loss [4,44]. Six input factors required for model application such as rainfall erosivity, soil erodibility, slope length and slope steepness, cover management, and conservation support practices (Figure 2) [52] were integrated using the model builder interface embedded in the ArcGIS software version 10.5 (Environmental Systems Research Institute (Esri), Inc., Redlands, CA, USA). Applying the nearest-neighbor method, all of the model factors derived from a multisource dataset with different spatial resolution were resampled to a 30 m × 30 m cell size and reprojected to a standard spatial reference system of World Geodetic System 1984 spheroid, Universal Transverse Mercator, and Adindan Zone 37 N. The RUSLE model equation is expressed as [54]:

$$A = R \times K \times L \times S \times C \times P \tag{2}$$

where A is average annual soil loss per unit area; R is the rainfall-runoff erosivity factor; K is a soil erodibility factor; LS is a slope length-steepness factor; C is a cover management factor; P is a support practice factor.

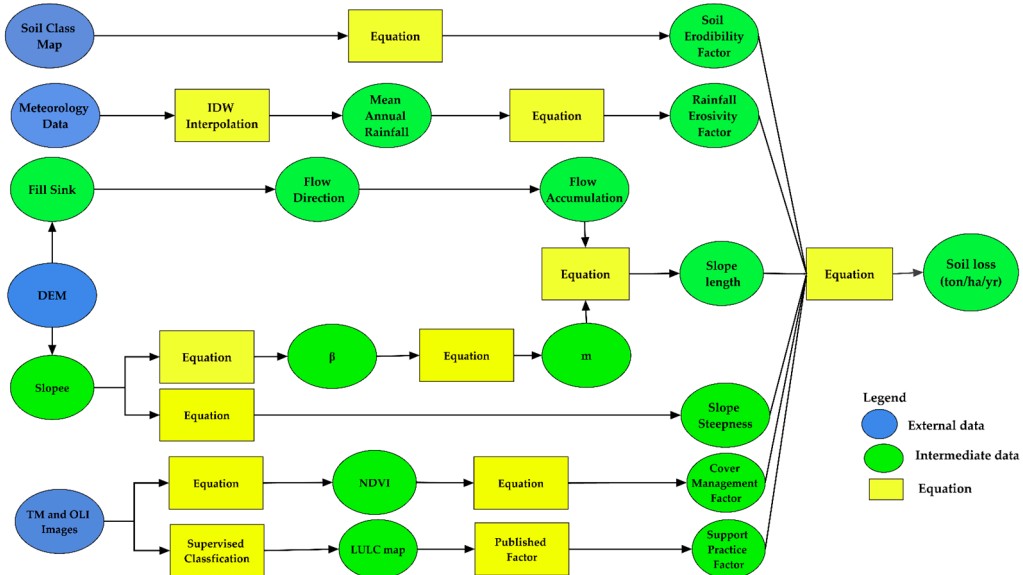

**Figure 2.** Flowchart for the soil loss estimation using the Revised Universal Soil Loss Equation (RUSLE) model framed in the ArcGIS model-builder interface.

Figure 2 shows an overall framework established in the ArcGIS environment for the integration of six input factors, derived from multisource spatial datasets, into the RUSLE model to estimate the soil loss rates in the Erer Sub-Basin. The model was run to estimate the actual annual rates of soil loss in the study landscape for the years 2000 and 2018. The soil erosion risk within the study area was classified into eight categories, based on previous work by Uddin et al. [5], and the estimated mean soil loss rates (t ha$^{-1}$ y$^{-1}$): very low (< 5), low (5–10), low medium (10–15), medium (15–20), high medium (20–25), high (25–35), very high (35–50), and extremely high (>50). Areas with a mean annual soil loss rates lower than low were rated as tolerable soil loss limit [52]. We created a cross-tabulated change detection matrix by overlaying the erosion risk maps pixel-by-pixel and calculated the percentage change, persistence, gain, loss, net-change, and a net-change-to-persistence ratio of erosion risk classes between 2000 and 2018. Moreover, SWC priority areas were identified and mapped based on the severity levels of soil erosion risk and a cross-tabulated matrix showing changes among erosion risk classes between the observed periods. Prioritization was done based on an MCDR method. We followed the methodological framework of Zhang et al. [88], who tested the capability of an MCDR approach in identifying priority areas to control soil erosion.

Rainfall and Runoff Erosivity Factor (R)

Rainfall-runoff erosivity is the primary factor causing soil erosion and accounts for about 80% of the soil loss [52,89]. The R factor is an index that reflects the capability of rainfall-runoff to detach and transport the soil particles that are experimentally determined by taking into consideration the intensity and a maximum duration of rainfall in a particular area of interest (Figure 3a) [52,87–92]. The R factor value was calculated based on mean yearly precipitation for the period 1998–2018, computed as the mean of total rainfall at fifteen local metrological stations distributed across the sub-basin, using the erosivity computation formula of Lo et al. [92]:

$$R = [38.46 + (3.48 \times P)] \tag{3}$$

where P is an annual average rainfall (mm).

Soil Erodibility Factor (K)

The soil erodibility factor, K, represents prolonged influences of soil profile characteristics and inherent soil properties on average soil loss measured on a standard plot condition [51,82,88,93]. The most important soil properties that affect soil erosion are soil organic matter content, soil texture, drainage ratio, and soil structure [5]. In this study, the K factor value was calculated based on the formula given by Wischmeier and Smith [52] using the FAO harmonized digital soil map [78], as follows (Equation (4)).

$$K = 2.1 \times 10^{-6} \times M^{1.14} \times (12 - OM) + 0.325 \times (P - 2) + 0.025 \times (S - 3) \tag{4}$$

where M = (percentage silt + percentage very fine sand) (100 percent clay); OM = the percentage of organic matter content; P = profile permeability; and S = structure classes.

The spatial distribution of the soil erodibility in the Erer Sub-Basin is shown in Figure 3b, with mean values ranging from 0.36 t h MJ $^{-1}$ mm$^{-1}$ to 0.42 t h MJ $^{-1}$ mm$^{-1}$ (Table S1). The lowest value for soil erodibility was obtained from the dystric cambisols (0.36 t h MJ $^{-1}$ mm$^{-1}$) which are typically found in the northwest of the study landscape. The most erodible soil classes included the eutric regosols (0.37 t h MJ $^{-1}$ mm$^{-1}$) and the eutric nitosols (0.42 t h MJ $^{-1}$ mm$^{-1}$) with a relatively higher sand content (> 68 percent) are situated in the north, northeast, and southwest of the study landscape.

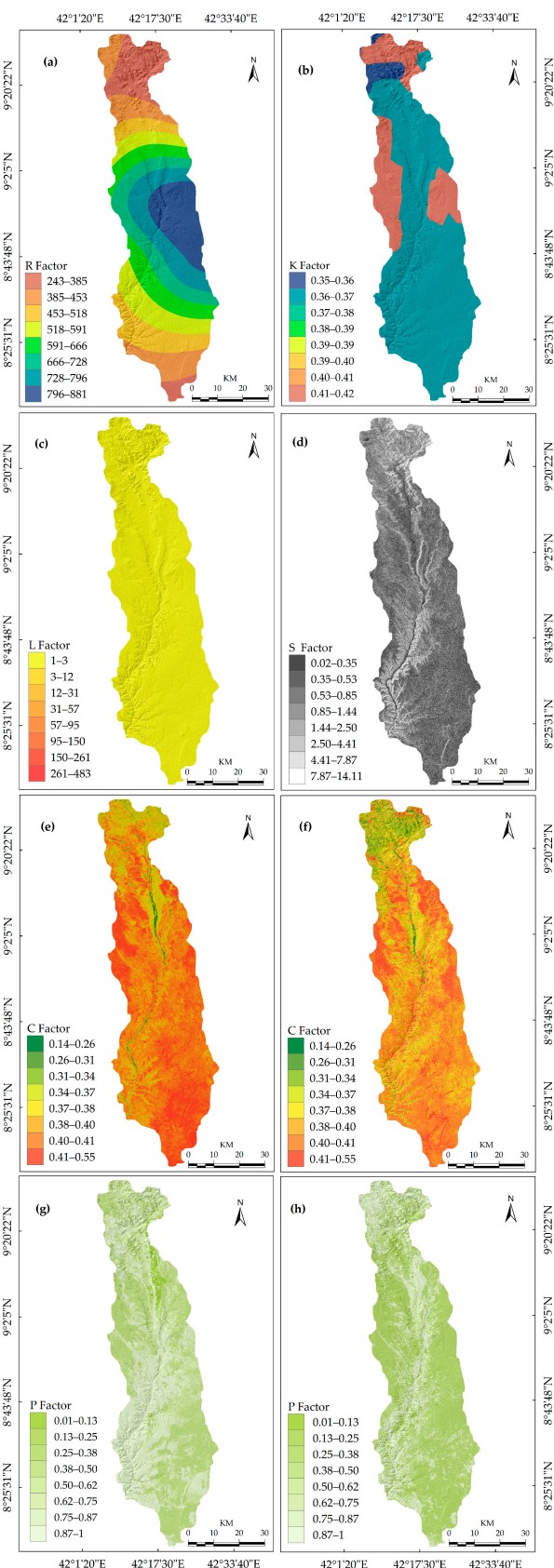

**Figure 3.** Rainfall-erosivity (R) factor (**a**); Soil erodibility (K) factor (**b**); Slope length (L) factor (**c**), Slope steepness (S) factor (**d**); Cover management (C) factor in 2000 (**e**) and 2018 (**f**); Support practice (P) factor in 2000 (**g**) and 2018 (**h**) in the Erer Sub-Basin, North East Shebelle Basin, Ethiopia.

Slope Length and Steepness (LS) Factor

The dimensionless slope length and steepness factor, LS, represent the effect of slope gradient on soil loss, can be determined as a product of the slope length (L) and slope steepness (S) [90,91]. The increase in slope length and slope steepness can cause a higher overland flow speed and runoff volume, which result in a high amount of soil loss [92]. The LS factor of the RUSLE model represents the proportion of soil loss on a given slope length and steepness to soil loss from a 22.13 m slope length and a steepness of 9% with all other conditions remains the same [52,93,94]. The L and S factors were calculated from a 30m resolution DEM image covering the sub-basin area using the following equations (Figure 3c,d).

$$L = \left(\frac{\lambda}{22.1}\right)m \tag{5}$$

where $\lambda$ is the horizontal field slope length in meters, and m is the variable slope length exponent calculated from the ratio of rill-to-interrill erosion slope steepness: 0.5 for slopes steeper than 4.5%; 0.4 for slopes between 3%−4.5%; 0.3 for slopes between 1%−3%, and 0.2 on slopes lower than 1%.

To represent the heterogeneity of slope steepness in the sub-basin area, the slope gradients were sub-divided into a number of segments by taking into consideration the unit upslope contributing areas [94–99].

$$L_{i,j} = \frac{\left(A_{i,j-in} + D^2\right)^{m+1} - A_{i,j-in}^{m+1}}{D^{m+1} \times x_{i,j}^m \times 22.13^m} \tag{6}$$

where $A_{i,j-in}$ is the contributing area at the inlet of the grid cell (i, j) is measured in $m^2$; $D$ is the grid cell size (meters); $x_{i,j}$ is $\sin a_{i,j} + \cos a_{i,j}$; $A_{i,j}$ is the aspect direction of the grid cell (i, j); and $m$ is the slope length exponent associated to the share of $\beta$ of rill-to-interrill erosion (Equations (7) and (8)) [95,98]:

$$m = \left(\frac{\beta}{1 + \beta}\right) \tag{7}$$

$$\beta = \frac{\frac{\sin \theta}{0.0896}}{\left[0.56 + 3 \times (\sin \theta)^{0.8}\right]} \tag{8}$$

$\theta$ is the slope steepness angle in degrees (Equation (9a,b)) [96,100].

$$S = 10.8 \sin \theta + 0.03, \text{ where slope gradient} < 9\% \tag{9a}$$

$$S = 16.8 \sin \theta - 0.50, \text{ where slope gradient} \geq 9\% \tag{9b}$$

Cover Management (C) Factor

The cover management factor, C, represents the proportion of soil loss from the field under a given crop management practices to that from clean-tilled continuous plowed land [101]. Following De Jong [102], the cover management factor was interpreted based on the Normalized Difference Vegetation Index (NDVI) generated using satellite images from Landsat 5 TM and Landsat 8 OLI sensors. The estimated C factor for 2000 and in 2018 is given in Figure 3e,f.

$$C = 0.431 - 0.805 \times NDVI \tag{10}$$

Support Practice (P) Factor

The P factor indicates the effects of various conservation practices in minimizing the amount and rate of soil loss owing to rainfall-runoff [96,103–107]. The value of the P factor is conventionally determined based on the types of soil conservation measures applied in a given area. Due to the constraints of field-based measurements concerning conservation practices put in place within the

study area, we determined the values of the P factor based on an alternative method recommended by Wischmeier and Smith (Table 1) [52]. For this purpose, the LULC maps interpreted from the Landsat satellite images and the slope map determined from the DEM were used to drive the spatial distribution maps of the P factor in 2000 and 2018 (Figure 3g,h).

**Table 1.** Conservation support practice (P) factor values [52].

| Land Use Type | Slope (%) | *p* Values |
|---|---|---|
| Agricultural land use | 0–5 | 0.1 |
| | 5–10 | 0.12 |
| | 10–20 | 0.14 |
| | 20–30 | 0.19 |
| | 30–50 | 0.25 |
| | 50–100 | 0.33 |
| Nonagricultural land use | 0–100 | 1.00 |

## 3. Results and Discussion

### 3.1. LULC Classfication

Six LULC classes identified in the Erer Sub-Basin were classified for the year 2000 and 2018, as shown in Figure 4. This includes bare land, cropland, forestland, settlement, shrubland, and water body, with a proportion of each LULC class in 2000 contributes 8.03%, 47.92%, 2.99%, 0.2%, 40.67%, and 0.18% of the total study area, respectively. Each LULC classes in 2018 accounts for 9.71%, 64.36%, 1.42%, 0.61%, 23.87%, and 0.03% of the total study area, respectively (Table 2). The classified LULC images illustrate that cropland was the most dominant LULC class in the study landscape in both 2000 and 2018, followed by shrubland and bare lands (Figure 4a,b).

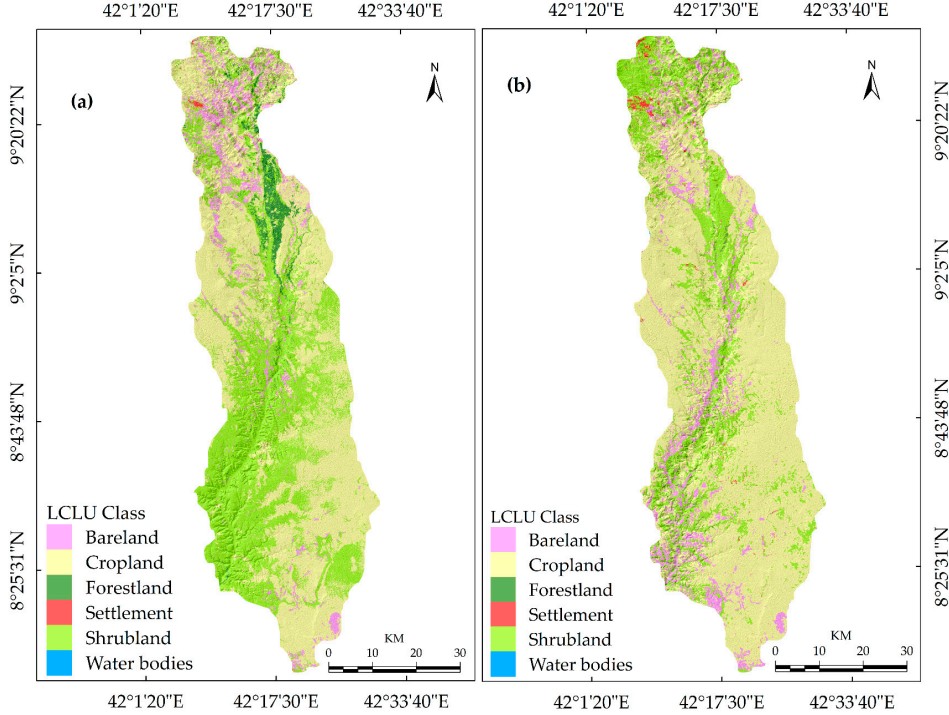

**Figure 4.** Land use and land cover (LULC) map of the Erer Sub-Basin, North East Shebelle Basin, Ethiopia; (**a**) 2000 and (**b**) 2018.

**Table 2.** Areal statistics of classified land use and land cover change (LULC) classes for 2000 and 2018.

| LULC Class | 2000 | | 2018 | |
|---|---|---|---|---|
| | Area (km$^2$) | % | Area (km$^2$) | % |
| Bare land | 310.00 | 8.03 | 374.81 | 9.71 |
| Cropland | 1849.70 | 47.92 | 2484.33 | 64.36 |
| Forestland | 115.42 | 2.99 | 54.81 | 1.42 |
| Settlement | 7.72 | 0.20 | 23.55 | 0.61 |
| Shrubland | 1569.88 | 40.67 | 921.39 | 23.87 |
| Water body | 7.33 | 0.18 | 1.16 | 0.03 |
| Total | 3860.05 | 100 | 3860.05 | 100 |

Presented in Table 3 is a statistical summary of classification accuracy assessment for the 2000 and 2018 LULC images. The user's accuracy, producer's accuracy, Kappa (Kˆ) coefficient, overall accuracy, overall Kˆ coefficient, commission and omission errors were utilized to validate the classification accuracies based on randomly generated reference points for LULC classes (Table 3). The diagonal values down an error matrix indicate reference samples that are accurately classified and off-diagonal entries are the misclassified references correspond to individual LULC classes. The overall classification accuracies attained based on the stratified random sampling method were 94.00% in 2000 and 96.33% in 2018, with a kappa coefficient of 0.93 and 0.95, respectively. The user's and producer's accuracies obtained per LULC classes ranged from 86.27% (bare land in 2000) to 100% (forestland, settlement, and water body in 2018) and, 80.00% (settlement in 2000) to 100% (shrubland in 2018), respectively. The bare land LULC class had a relatively high commission error, while the settlement had a relatively high omission error both in the 2000 and 2018 classified images. Thus, the classified satellite images have overestimated the bare land and underestimated settlement area. This is probably due to the spectral similarity of the bare land and the settlement LULC classes. From the statistical results of the classification accuracy assessment presented in Table 3, it was confirmed that the classified images agree with the training samples, and is, therefore, satisfactory to conduct a change detection analysis [81,82].

### 3.2. Assessment of LULCC in the Erer Sub-Basin

LULCC is closely related to human decisions and complex interactions among multiple activities, working at a location [108,109]. Up-to-date information about the dynamics of LULCC and its drivers is an increasingly important issue in the examination of environmental change for identifying the current resource situation and designing sustainable resource management measures [108–110]. The present study found a considerable LULCC in the Erer Sub-Basin, which is located in the Upper Wabi Shebelle Basin. The extent of changes varied among the LULC classes during the period between 2000 and 2018. During the study period, areas covered by forestland, shrubland, water body showed a considerable reduction (Table 4). The forestland converted during the period of the assessment totaled 60.60 km$^2$, which is about 2.99% of the total area that covered in 2000. Likewise, shrub land cover has decreased in the study landscape by 41.31% of the total area. Water body also showed a reduction of 84.21% of the total area. The decline in a water body is probably due to the expansion of settlement and cropland in shrubland and forestland in the study landscape. On the contrary, bare land, cropland, and settlement LULC classes have increased by 20.9%, 34.31%, and 205%, respectively.

**Table 3.** Accuracy statistics for the classified LULC maps in percent.

| Years/ Class Name | Classification Accuracy | | | | | | | | |
|---|---|---|---|---|---|---|---|---|---|
| | Bare Land | Cropland | Forestland | Settlement | Shrub Land | Water Body | Raw Total | User's Accuracy | Commission Error |
| 2000 | | | | | | | | | |
| Bare land | *44* | 0 | 0 | 7 | 0 | 0 | 51 | 86.27 | 13.73 |
| Cropland | 0 | *80* | 0 | 0 | 0 | 0 | 80 | 100 | 0 |
| Forestland | 0 | 0 | *45* | 0 | 5 | 0 | 50 | 90.00 | 10.00 |
| Settlement | 0 | 0 | 0 | *28* | 0 | 0 | 28 | 100 | 0 |
| Shrubland | 6 | 0 | 0 | 0 | *65* | 0 | 71 | 91.55 | 8.45 |
| Water body | 0 | 0 | 0 | 0 | 0 | *20* | 20 | 100 | 0 |
| Colum Total | 50 | 80 | 45 | 35 | 70 | 20 | 300 | | |
| Kˆ statistics | 0.84 | 1.00 | 0.88 | 1.00 | 0.89 | 1 | | | |
| Producer's Accuracy | 88 | 100 | 100 | 80 | 92.86 | 100 | Overall Accuracy = 94 Overall Kˆ = 0.93 | | |
| Omission Error | 12.00 | 0 | 0 | 20 | 7.14 | 0 | | | |
| 2018 Bare land | *48* | 2 | 0 | 2 | 0 | 0 | 52 | 92.31 | 7.69 |
| Cropland | 3 | *74* | 0 | 1 | 0 | 0 | 81 | 96.30 | 3.70 |
| Forestland | 0 | 0 | *43* | 0 | 0 | 0 | 43 | 100 | 0 |
| Settlement | 0 | 0 | 0 | *32* | 0 | 0 | 32 | 100 | 0 |
| Shrubland | 0 | 0 | 2 | 0 | *70* | 2 | 74 | 94.59 | 5.41 |
| Water body | 0 | 0 | 0 | 0 | 0 | *18* | 18 | 100 | 0 |
| Colum Total | 51 | 76 | 45 | 35 | 70 | 1 | 300 | | |
| Kˆ statistics | 0.91 | 0.95 | 1.00 | 1.00 | 0.93 | 1 | | | |
| Producer's Accuracy | 96 | 97.5 | 95.56 | 91.43 | 100 | 90 | Overall Accuracy = 96.33 Overall Kˆ = 0.95 | | |
| Omission Error | 4 | 2.50 | 4.44 | 8.57 | 0 | 10 | | | |

**Table 4.** Temporal change in the spatial extent of LULC classes in percentage (%).

| LULC Class | Rate of Changes (2000–2018) | |
|---|---|---|
| | Area (km²) | % |
| Bare land | 64.81 | 20.91 |
| Cropland | 634.63 | 34.31 |
| Forestland | −60.60 | −52.51 |
| Settlement | 15.83 | 205.00 |
| Shrubland | −648.49 | −41.31 |
| Water body | −6.18 | −84.21 |

Changes between LULC classes in the period 2000–2018 are provided in Table 5. The loss column represents the amount of LULC that experienced a gross loss of category *i* between 2000 and 2018, while the gain row indicates the LULC that experienced a gross gain of class *j* between the same periods. The change detection matrix shows that overall, nearly 43.48% of the land within the study landscape experienced LULCC during the period between 2000 and 2018. As shown in Table 5, the major LULCCs identified during the study period were from shrubland to cropland (21.26% of the original shrubland has been converted to cropland), cropland to shrubland (6.99% of the original cropland has been converted to shrubland), shrubland to bare land (4.74% of the original shrubland has been converted to bare land), and forestland to shrubland (2.07% of the original forest has been converted to shrubland).

**Table 5.** Change matrix showing the LULC classes changes between 2000 and 2018 in percentage (%).

| LULC Class | Bare land | Cropland | Forestland | Settlement | Shrubland | Water Body | 2000 |
|---|---|---|---|---|---|---|---|
| Bare land | *3.33* | 3.48 | 0.01 | 0.14 | 1.07 | 0.00 | 8.03 |
| Cropland | 1.55 | *39.04* | 0.18 | 0.16 | 6.99 | 0.01 | 47.92 |
| Forestland | 0.05 | 0.53 | *0.33* | 0.01 | 2.07 | 0.00 | 2.99 |
| Settlement | 0.03 | 0.04 | 0.00 | *0.07* | 0.06 | 0.00 | 0.20 |
| Shrubland | 4.74 | 21.26 | 0.80 | 0.22 | *13.65* | 0.00 | 40.67 |
| Water body | 0.01 | 0.01 | 0.10 | 0.01 | 0.03 | *0.02* | 0.19 |
| Summary | | | | | | | *56.52* |
| 2018 | 9.71 | 64.36 | 1.42 | 0.61 | 23.87 | 0.03 | |

During 2000 and 2018, about 6.49% of the LULCC was occurred due to swap change, wherein a comparable area was gained and lost among the LULC classes (Table 6). During the study period, the persistence of the LULC classes accounts for 56.52% of the total area. The change analysis results generally indicate that the cropland and the shrubland were relatively the highest persistence LULC classes, whereas the water body was the lowest-persistence class. Out of the 47.92% and 40.67%, the cropland and the shrubland LULC classes covered in 2000 around 39.04% and 13.65% of the total area remained unchanged in 2018, and the remaining 8.88% and 27.02% were converted to other LULC classes, respectively. Similarly, the cropland showed the highest gross gain (25.32%) due to the area mainly converted from shrubland, bare land, and forestland. Although the cropland gained areas converted from shrubland, bare land, and forestland, it experienced a net loss of about 8.90 % of the total area. The shrubland experienced the highest net loss among the LULC classes. It accounts for about 27.02% of the total area (with about 21.26%, 4.74%, 0.80%, and 0.22% of shrub land swapped into cropland, bare land, forestland, and settlement, respectively), whereas about 160.44 km$^2$ of new shrub land was established at the expense of cropland (6.99%), forestland (2.07%), bare land (1.07%), and water body (0.03%). The net change-to-persistence ratio was relatively higher for settlement, cropland, and bare land, showing their persistence in comparison to their net loss. On the contrary, the net change-to-persistence ratio was negative for a water body, forestland, and shrubland, suggesting their net loss rather than their persistence in the study landscape [85]. The findings of this study were consistent with numerous studies' findings in other parts in Ethiopia [111–119], and elsewhere in the world [5,16,17]. These studies have revealed a heterogeneity in the spatial and temporal extent of LULCCs.

**Table 6.** LULCCs in the period 2000–2018 in percent.

| LULC Class | Persistence | Gain | Loss | Total Change | SWAP | Absolute Value of Net Change | Gain to Persistence | Loss to Persistence | Net Change to Persistence |
|---|---|---|---|---|---|---|---|---|---|
| Bare land | 3.34 | 6.38 | 4.7 | 11.08 | 9.4 | 1.68 | 1.92 | 1.41 | 0.50 |
| Cropland | 39.04 | 25.32 | 8.88 | 34.2 | 17.76 | 16.44 | 0.65 | 0.23 | 0.42 |
| Forestland | 0.33 | 1.09 | 2.66 | 3.75 | 2.18 | 1.57 | 3.30 | 8.06 | −4.76 |
| Settlement | 0.07 | 0.54 | 0.13 | 0.67 | 0.26 | 0.41 | 7.71 | 1.86 | 5.86 |
| Shrubland | 13.65 | 10.22 | 27.02 | 37.24 | 20.44 | 16.8 | 0.75 | 1.98 | −1.23 |
| Water body | 0.02 | 0.01 | 0.16 | 0.16 | 0.00 | 0.15 | 0.50 | 8.00 | −7.50 |
| Total | 56.45 | 43.55 | 43.55 | 43.55 | 6.49 | 37.06 | | | |

For example, Kindu et al. [111] found cropland expansion at the expense of woodlands, forest, and grassland in Munessa-Shashemene landscape of the Ethiopian highlands. Similar results were also found in the south-central Ethiopia, where agriculture land expansion has reached its peak on the suitable land over the period 1972–2013 and continued to occupy marginal lands affecting the forest biodiversity [112]. In accordance with the findings of the current study, Mengistu et al. [113] reported an increase in cropland while a downward trend in riverine trees and shrub-grassland in the Upper Dijo River Catchment of the south-central Ethiopia. Supporting our findings, another study investigated the dynamics of LULCC and the woody vegetation diversity in the human-driven landscape of the Gilgel Tekeze Catchment reported an increase in cropland and settlement area while a decrease in the forest and the bushland [114].

On their part, Fetene et al. [115] and Belay et al. [116] analyzed the LULCCs in the Awash National Park (ANP) and the Nech-Sar National Park (NSNP) in Ethiopia. Their studies' findings showed that that the main drivers of LULCCs occurred within the two national parks–causing enormous destructions in wildlife habitat–has been attributed to changes in the land tenure system and regime changes, immigration, drought, poaching, and deforestation in combination with ever-increased pressure from the local community and livestock [115,116]. A recent report by Hailemariam et al. [117] also concluded that population growth resulted in a high demand for cropland expansion, which in turn, has triggered a decrease in the areas of forest cover, shrubland, and grassland in the Bale mountain eco-region of Ethiopia. In a related study conducted in the Gilgel Tekeze Catchment of the northern Ethiopia highlands, Haregeweyn et al. [118] suggested an integrated catchment management measures to minimize the adverse impacts of LULCC on sustainable hydrological system. Tadesse et al. [119], in contrast, reported a regeneration of vegetation cover in the Yezat Watershed of northwestern Ethiopia, which was attributed to an integrated watershed management practices taken over the period between 2010 and 2015.

### 3.3. Overview of Soil Erosion in the Erer Sub-Basin

The spatial distribution of soil erosion risks in the Erer Sub-Basin is shown Figure 5a in 2000 and Figure 5b in 2018, while the estimated soil loss rates and the erosion risk classes are provided in Table 7. The estimated total annual actual soil loss in the study landscape was 1.01 million tons in 2000 and 1.52 million tons in 2018. Our estimate of soil loss falls within the range of the previous findings that estimated the soil loss rate in the highland areas of Ethiopia from 1248 to 23,400 million tons [30]. The soil erosion risk had shown a high spatial variation across the study landscape (Figure 5). As it can be observed from Figure 5, high soil erosion risk areas were located in the northeast, southwest, and the central parts of the Erer Sub-Basin, which are also, found in the rugged topography and steep slopes. Relatively less eroded areas were situated in the lower elevations in the eastern and western parts of the sub-basin, where the slope inclination is ranging from nearly zero to ten percent. Similar results have been reported by the earlier studies that attributed lower soil loss rate to gentle slopes while a higher soil loss in steep slope areas [18,120–123].

The mean annual soil loss rate was estimated at 75.85 t ha$^{-1}$ y$^{-1}$, 107.07 t ha$^{-1}$ y$^{-1}$, in 2000 and 2018, respectively, for the entire sub-basin. It was also found in this study that the mean soil loss of 2018 increased by an average of 41.16 t ha$^{-1}$ y$^{-1}$ when compared to the mean soil loss of 2000. The estimated mean annual soil loss rate in the present study area is considerably higher than that of the maximum tolerable soil loss limits estimated for the agro-ecological regions (18 t ha$^{-1}$ y$^{-1}$) [124] and soil formation rates for the various land units in Ethiopia [125], and to the normal soil loss tolerances indicated by the Wischmeier and Smith (5–11 t ha$^{-1}$ y$^{-1}$) [52]. The estimated mean rate of soil loss is also higher than the findings of previous investigators in the Upper Wabi Shebelle Basin [18,67,68,71], and other river basins in Ethiopia [126,127]. On the contrary, the estimated soil erosion rates are much lower than the local scale studies that estimated the soil loss rate of 935 t ha$^{-1}$ y$^{-1}$ in the Beshillo Catchment of the Blue Nile Basin [128]; 243 t ha$^{-1}$ y$^{-1}$ in northwestern highlands Ethiopia [129], and 321 t ha$^{-1}$ y$^{-1}$ in the eastern escarpment of Wollo [130].

According to the estimated rates of mean annual soil loss, the erosion risk was classified into eight classes extending from the very low to extremely high. The proportion of the area at very low risk covered a larger part of the sub-basin area (Table 7) accounts for about 48.87% and 46.22% of the total study area in 2000 and 2018, respectively. The area at very low and extremely high risk of soil erosion went down from 48.87% and 2.36% in 2000 to 46.22% and 2.14% respectively, in 2018. On the contrary, the low, low medium, medium, high medium, high, and very high have increased by 6.54%, 1.62%, 2.87%, 10.76%, 16.55%, and 16.55% of the total study area, respectively. Areas with a mean annual soil loss greater than low have increased by 3.80 % of the total study area. The results indicate that the estimated erosion rate for about 23.98% of the sub-basin area exceeds the maximum tolerable soil erosion threshold [52].

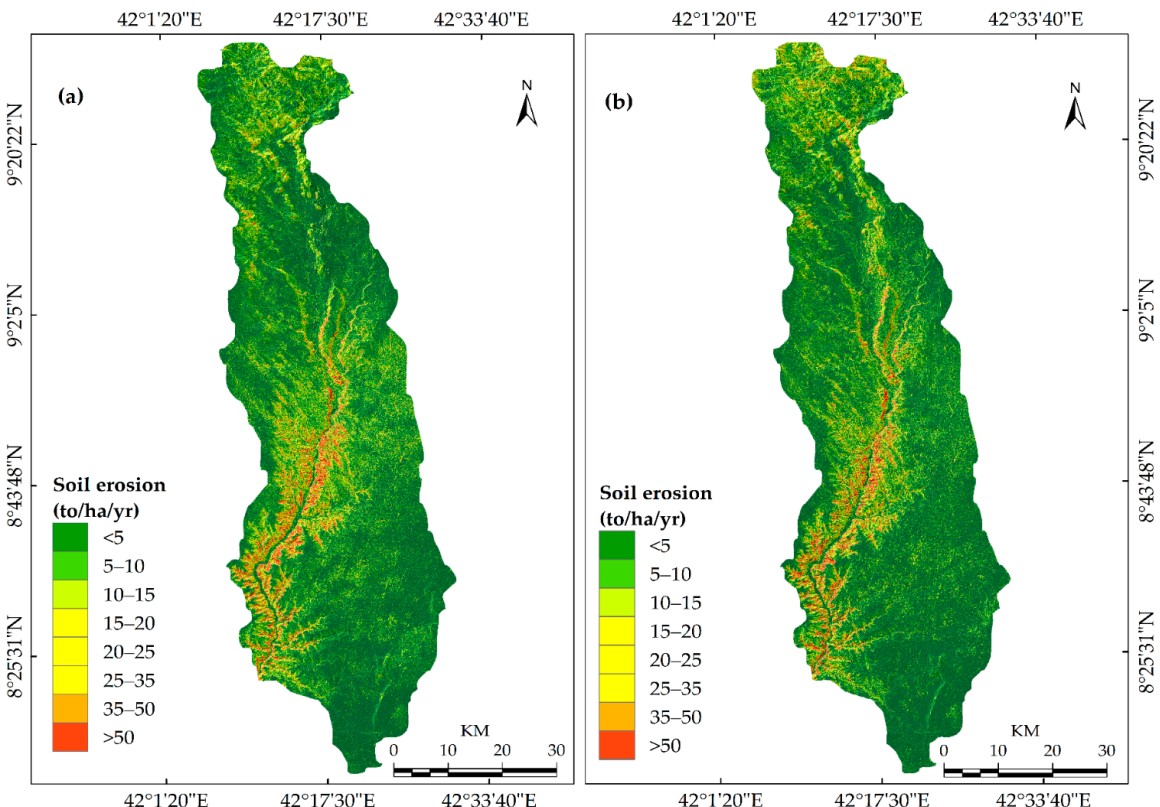

**Figure 5.** Soil erosion risk in the Erer Sub-Basin, North East Shebelle Basin, Ethiopia; (**a**) in 2000, (**b**) 2018.

**Table 7.** Areas (km²), percentages, and changes in soil erosion risk classes between 2000 and 2018.

| Erosion Risk Class | Soil Loss (t ha⁻¹ y⁻¹) | 2000 | | 2018 | | Rate of Changes (2000–2018) | |
|---|---|---|---|---|---|---|---|
| | | Area (km²) | % | Area (km²) | % | Area (km²) | % |
| Very low | <5 | 1883.49 | 48.87 | 1783.50 | 46.22 | −99.99 | −5.31 |
| Low | 5–10 | 1078.90 | 27.99 | 1149.43 | 29.79 | 70.53 | 6.54 |
| Low medium | 10–15 | 399.01 | 10.35 | 405.46 | 10.51 | 6.45 | 1.62 |
| Medium | 15–20 | 190.96 | 4.95 | 196.45 | 5.09 | 5.49 | 2.87 |
| High medium | 20–25 | 104.22 | 2.70 | 115.43 | 2.99 | 11.21 | 10.76 |
| High | 25–35 | 63.98 | 1.66 | 74.57 | 1.93 | 10.59 | 16.55 |
| Very high | 35–50 | 42.51 | 1.10 | 50.98 | 1.32 | 8.47 | 19.92 |
| Extremely high | >50 | 90.97 | 2.36 | 82.60 | 2.14 | −8.37 | −9.20 |

Table 8 presents the estimated soil loss from each LULC class in 2000 and 2018. The estimated mean soil loss increased for LULC classes during the period between 2000 and 2018, showing that the LULCC have detrimental impacts on soil loss by water erosion [4,16,17]. Understanding the dynamics in LULCCs and consequent changes in the distribution of soil erosion risk can provide a spatial decision support tool for conservation planners to develop an appropriate SWC measures. Settlement area that occupied about 0.20%, 0.61% of the sub-basin area, in 2000 and 2018, accounted for 2.64% and 1.99% of the total soil loss, respectively. During 2000 and 2018, the minimum amount of soil loss was estimated in water bodies, with a mean erosion rate of 0.02 t ha⁻¹ yr⁻¹ and 0.26 t ha⁻¹ yr⁻¹, respectively. The soil loss from the water body, forestland, and settlement was relatively low, and the annual soil loss from cropland was accounted for 42.06% and 48.34% of the soil erosion in 2000 and 2018, respectively.

**Table 8.** Mean soil loss rate with respect LULC classes in the Erer Sub-Basin.

| LULC Class | 2000 | 2018 |
|---|---|---|
| | Mean Soil Loss (t ha$^{-1}$ yr$^{-1}$) | Mean Soil Loss (t ha$^{-1}$ yr$^{-1}$) |
| Bare land | 8.98 | 15.78 |
| Cropland | 25.73 | 37.60 |
| Forestland | 0.02 | 2.47 |
| Settlement | 0.18 | 0.55 |
| Shrubland | 10.19 | 11.62 |
| Water body | 0.02 | 0.26 |

The cropland, bare land, and settlement had become the main causes of soil erosion in the study landscape, as the estimated mean soil loss rate for the three LULC classes have increased 11.88, 6.80, and 2.44 t ha$^{-1}$ yr$^{-1}$; however, their rates of changes varied 34.31%, 20.91%, and 205%, respectively.

Our findings coincide with those of the recent study by Yesuph and Dagnew [128] who showed that the cropland under a mono-cropping and intensive cultivation in the upslope areas were responsible for severe soil erosion in the Beshillo Catchment of the Blue Nile Basin. Validating the present study's findings, Belayneh et al. [120] also pointed out that cultivated land with a mean erosion rate of 45.68 t ha$^{-1}$ yr$^{-1}$ accounted for 62.06% of the total soil loss from the Gumara Watershed of the northwestern Ethiopia highland. The landscape that had experienced the LULCC during the period of the assessment accounted for about 43.48% of the total study area, of which about 11.44% revealed an increase in the estimates of soil loss of 75.66 t ha$^{-1}$ yr$^{-1}$. The remaining landscape under LULCC had undergone a decrease in actual soil loss of 116.63 t ha$^{-1}$ yr$^{-1}$. Of the landscape under LULCC experienced, a high in an estimated soil erosion rate corresponds to the area where the water bodies were changed to shrubland (increase in actual soil loss was 9.69 t ha$^{-1}$ yr$^{-1}$). Figure 6 shows that the second and the third detrimental LULCCs accounted for an increase in the actual soil erosion in the study area were conversions from forestland to shrubland (+9.51 t ha$^{-1}$ yr$^{-1}$) and from water bodies to bare land (+9.39 t ha$^{-1}$ yr$^{-1}$). At the same period, changes from forestland to bare land and settlement accounted for an increase in soil loss of 8.54 t ha$^{-1}$ yr$^{-1}$ and 7.02 t ha$^{-1}$ yr$^{-1}$, respectively. At the sub-basin level, the positive LULCCs that contributed to a significant reduction in the estimates of soil erosion were a change from shrubland to forestland and water body (Figure 6).

Table 9 shows the proportion of soil erosion risk classes change between 2000 and 2018. The diagonal of the transition matrix indicates the proportion of erosion risk classes that remained unchanged during the study period, while the off-diagonal elements account of a conversion from one class to other classes of soil erosion risk. The loss and gain row represent the percentage loss and gain in each erosion risk class, respectively. The change analysis results show that about 65.80% of the total erosion risk areas occupied in 2000 remained unchanged in 2018. The overall gain and loss of the soil erosion risk classes account for 34.21% and 34.18% of the total area, respectively. The highest net gain (12.64% of the total area) and gross loss (10.84% of the total area) was estimated in an erosion risk class of low. It accounts for about 0.83% of the total study area. The highest net-change (1.8% of the total area) and net-change-to-persistence ratio (2% of the total area) was estimated in the in the area at low and very high risk of erosion. The change analysis results indicate that the erosion risk areas increased by 8.28% of the total study area, and decreased by 5.93%, which reveals that the overall erosion risk condition is deteriorating in the study landscape. The present study's findings agree with those of the recent study by Weldemariam et al. [18] who indicated that the situation of soil loss risk in the Gobele Watershed has been worsening due to increases in the proportion of erosion risk areas by 19.67% of the total watershed area between 2000 and 2016. Uddin et al. [5], in contrast, found improvement in the situation of soil erosion in Nepal, where the mean soil loss rates have decreased from 8.76 t ha$^{-1}$ y$^{-1}$ in 1990-to 7.49 t ha$^{-1}$ y$^{-1}$ in 2010. Validating these findings, Jiu et al. [131] stated that an increase of water level and river surface and afforestation measures taken in the period 2000–2015 significantly reduced

the soil erosion risk in the Three Gorges Reservoir Region (TGRR), China. According to Jiu et al. [131], the interactions between NDVI and urbanisation as well as vegetation diversity and urbanisation are key factors influencing soil loss in the TGRR.

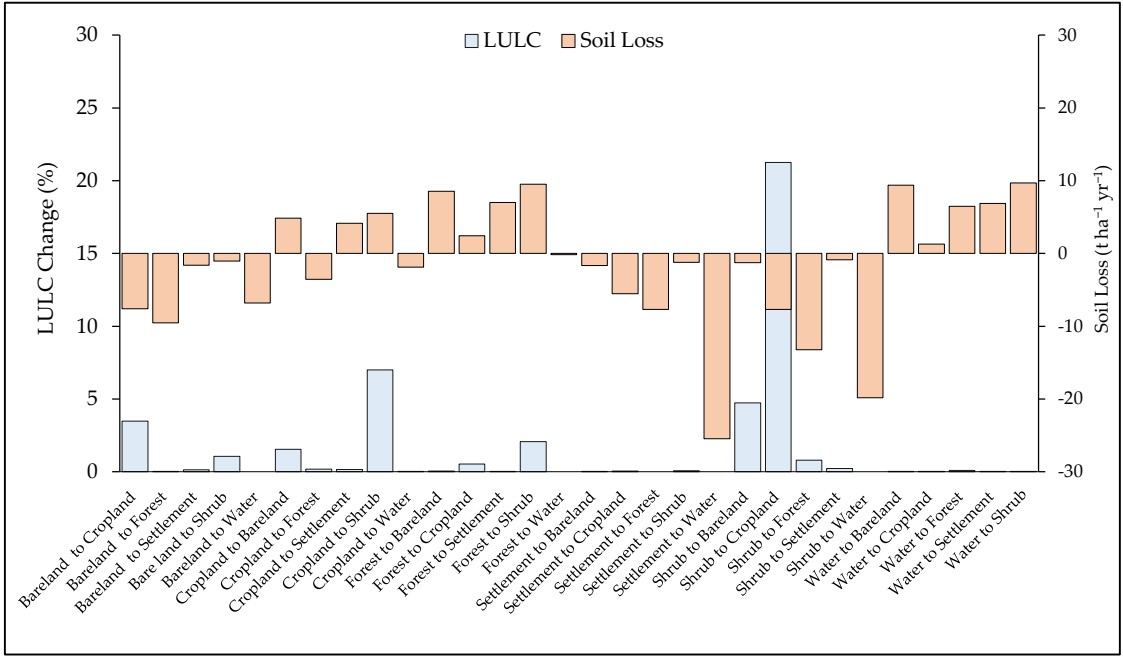

**Figure 6.** Land Cover Changes and their effects on soil erosion risk in the Erer Sub-Basin, North East Shebelle Basin, Ethiopia.

**Table 9.** Change of erosion risk classes between 2000 and 2018.

| Soil Erosion Risk Class | Very Low | Low | Low Medium | Medium | High Medium | High | Very High | Extremely High | Total 2000 | Loss |
|---|---|---|---|---|---|---|---|---|---|---|
| Very low | *40.55* | 7.99 | 0.16 | 0.07 | 0.06 | 0.04 | 0.00 | 0.00 | 48.87 | 8.32 |
| Low | 4.50 | *17.1* | 4.05 | 2.08 | 0.12 | 0.01 | 0.03 | 0.05 | 27.99 | 10.84 |
| Low-medium | 0.61 | 3.14 | *4.31* | 0.18 | 1.21 | 0.79 | 0.11 | 0.00 | 10.35 | 6.04 |
| Medium | 0.22 | 0.93 | 1.38 | *1.52* | 0.02 | 0.04 | 0.43 | 0.42 | 4.95 | 3.43 |
| High-medium | 0.14 | 0.48 | 0.09 | 1.06 | *0.62* | 0.01 | 0.00 | 0.30 | 2.70 | 2.08 |
| High | 0.09 | 0.05 | 0.26 | 0.05 | 0.85 | *0.27* | 0.00 | 0.08 | 1.66 | 1.39 |
| Very high | 0.05 | 0.00 | 0.17 | 0.01 | 0.07 | 0.65 | *0.11* | 0.02 | 1.10 | 0.99 |
| Extremely high | 0.03 | 0.06 | 0.09 | 0.12 | 0.04 | 0.13 | 0.63 | *1.27* | 2.36 | 1.09 |
| Summary | | | | | | | | | *65.80* | |
| Total 2018 | 46.20 | 29.79 | 10.52 | 5.10 | 2.99 | 1.93 | 1.32 | 2.15 | | |
| Gain | 5.65 | 12.64 | 6.20 | 3.58 | 2.38 | 1.67 | 1.21 | 0.88 | | |
| Net change | −2.67 | 1.80 | 0.17 | 0.14 | 0.28 | 0.28 | 0.22 | −0.21 | | |
| NP | −0.07 | 0.10 | 0.04 | 0.09 | 0.45 | 1.04 | 2.00 | −0.17 | | |

Overall persistence (i.e., the sum of the diagonals denotes the proportion of unchanged classes account for the total area). Net change = gain − loss in percent. Np denotes a net change-to-persistence ratio (i.e., net change/diagonals of each class).

## 3.4. Determination Conservation Priority Levels

Several previous studies highlighted the positive outcome of SWC measures for mitigating erosion risk, restoration of the degraded land while improving the soil fertility and land productivity [5,35–43]. The design and implementation of SWC measures need a spatially intrinsic information on soil loss and severity levels erosion risk [18,47,99]. In view of the fact that the distributions of soil erosion risk have shown a spatial variation within the sub-basin, we identified and mapped areas with a higher soil erosion rate as priority areas for SWC measures using an MCDR method (Figure 7) [88]. Determination of conservation priorities was done based on the estimated soil erosion rates and the cross-tabulated change detection matrix of erosion risk classes changes between 2000 and 2018. The portion of the

sub-basin area with high soil loss and increases in erosion risk grades were delineated in uppermost conservation priority levels (Table 10).

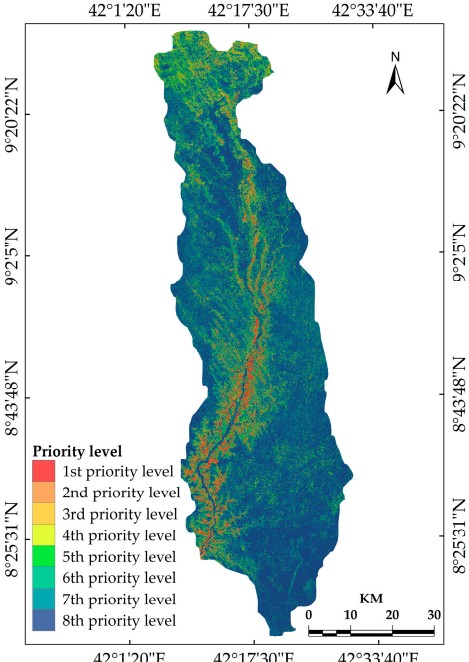

**Figure 7.** Conservation Priority levels of the Erer Sub-Basin.

**Table 10.** Area of the conservation priority level of the study area.

| Priority Level | Area (km$^2$) | Percentage (%) |
|---|---|---|
| 1st priority level | 96.55 | 2.50 |
| 2nd priority level | 92.00 | 2.38 |
| 3rd priority level | 82.73 | 2.14 |
| 4th priority level | 139.87 | 3.62 |
| 5th priority level | 209.84 | 5.44 |
| 6th priority level | 444.13 | 11.51 |
| 7th priority level | 903.31 | 23.40 |
| 8th priority level | 1891.63 | 49.01 |

Eight SWC priority areas were identified at the sub-basin scale revealed that the top three priority levels delineated for urgent SWC measures represent those areas within a higher soil loss rate and the large increase in erosion risk levels, with an area of 271.28 km$^2$ and accounts for 7.03 % of the sub-basin area. About 80.46% of the top three priority areas are situated in the Gursum, Babile, Fedis, Fik, and Gola Oda districts (Table S2), which are, located in the north, northeast, southwest, south, and south-west of the sub-basin. The remaining patches within these priority levels account for 19.54% of the total area, which are found in the upland within the Haramaya, Jarso, and Kombolcha districts, and the Harari Region that is located in the northern part of the sub-basin. The fourth-, fifth-, sixth-, seventh- and eighth-priority levels accounting for about 92.98% of the total study area need of negligible conservation measure to control soil loss and erosion risk.

## 4. Conclusions

Understanding the magnitude of LULCC and consequent changes in the spatial extent of soil erosion risk for the Erer Sub-Basin is the main aim of this study. The LULCC was examined based on multispectral Landsat satellite images acquired in 2000 and 2018. The soil erosion rate was estimated using the RUSLE model developed in the ArcGIS environment. According to our analysis, overall,

nearly 43.48% of the land in the study area experienced LULCCs in the 18 years (2000–2018) study period. During the study period, cropland, bare land, and settlement increased from 47.92%, 8.03% and 0.20% in 2000 to 64.36%, 9.71% and 0.61%, respectively, in 2018. On the contrary, areas covered by forestland, shrubland, and water body have decreased from 2.99%, 40.67% and 0.18% to 1.42%, 23.87%, 0.03%, respectively, in 2018. The change analysis matrix showed that cropland gained 25.33%, while shrubland lost 27.02% of the total area. The bare land and cropland expansion were found to be the major drivers of LULCC contributing to high soil loss rates, wherein the entire study area, an estimated total of 1.5 million tons of soil was displaced in 2018, of which 48.34% and 36.01 is lost from cropland and bare land, respectively. The findings of the study generally elucidate that the LULCC have a detrimental impact on soil erosion. Mean soil loss rate increased from 75.85 t ha$^{-1}$ y$^{-1}$ in 2000 to 107.07 t ha$^{-1}$ y$^{-1}$ in 2018, with high erosion risk areas being in the central, northeastern, and southwestern Erer Sub-Basin. Based on the estimated rate of mean annual soil loss, erosion risk was classified into eight classes, showing that over one-third of the study landscape (76.01%) was estimated to have erosion risk below low medium with a man soil loss lower than 10 t ha$^{-1}$ y$^{-1}$. The erosion risk that experienced changes during the study period accounts for about 34.2% of the total study area, of which about 15.93% decreased and 18.28% showed an increase in the study landscape. This shows that the erosion risk condition is deteriorating in the study landscape. The study area was classified into eight SWC priority levels based on the severity levels of erosion risk. About 7.02% of the sub-basin area was found to be under the first-, second-, and third-priority levels that need intense SWC measures. Further detailed investigations based on data from primary and secondary sources would be important in identifying driving socioeconomic forces and consequences of LULCCs and suggest possible alternative options to establish sustainable resource management practices in the study area.

**Supplementary Materials:** The following are available online at http://www.mdpi.com/2073-445X/9/4/111/s1, Table S1: Attributes of soil units and calculated soil erodibility (K) factor. Table S2: List of priority districts identified for SWC planning in the Erer Sub-Basin.

**Author Contributions:** G.W.W. conceived and designed the method, performed the experiment, and drafted the manuscript. A.E.H. performed the experiment, reviewed, and commented on the manuscript. All authors have read and agreed to the published version of the manuscript.

**Funding:** This research received no external funding.

**Acknowledgments:** The authors would like to thank the three anonymous reviewers and the editors for their valuable comments for improving this article.

**Conflicts of Interest:** The authors declare no conflict of interest.

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
