# Peer review of "Effect of Land Use and Land Cover Change on Soil Erosion in Erer Sub-Basin, Northeast Wabi Shebelle Basin, Ethiopia"

_land, doi:10.3390/land9040111_

Round 1

Reviewer 1 Report

It is an excellent research but poorly explained research work. The title of the manuscript suggest that the research will establish a relationship between LULC change and soil erosion. However, I could not found that relationship explained in the manuscript. It is suggested that the authors should identify LULC changes and related those areas of change with the soil erosion risk (a relationship between figure 4 and figure 5). The manuscript cannot be recommended for publishing without establishing this relationship.

Data acquisition dates of satellite data used in this study has not been provided. Similarly, band combination used for LULC classification has not been mentioned at all. This could be as crucial information for any study. Please provide this information.

A through revision is required for better explanation of all the areas of manuscript. I have also added few comments to the attached manuscript for possible improvements.

Author Response

Reviewer #1:

General comments:

It is an excellent research but poorly explained research work. The title of the manuscript suggests that the research will establish a relationship between LULC change and soil erosion. However, I could not found that relationship explained in the manuscript. It is suggested that the authors should identify LULC changes and related those areas of change with the soil erosion risk (a relationship between figure 4 and figure 5). The manuscript cannot be recommended for publishing without establishing this relationship.

[Response 1]: Many thanks to the reviewer 1 for his/her constructive comments and providing such detailed and helpful comments, which helped to enrich the quality of the manuscript. As per the comments, we have discussed the LULC changes detected between the study periods and related those areas of change with the soil erosion risk (Line: 165-185). We have added a new figure (Figure 6), so that a relationship between LULC change and soil erosion risk is established. We have further checked the attached document ‘peer-review-4610955.v1.pdf’ and incorporated all the comments in the revised version of the manuscript. Please see our detailed responses below.

Specific comments:

Please explain how population growth intensify soil erosion?

[Response 2]: Apologise for the confusion caused. We have revised the sentence to improve clarity (Line 63-64). The sentence has now been rewritten as “Rapid population increase and growing demand posed greater pressure on land resources, leading to severe soil erosion and land degradation in many parts of the country”.

Introduction: Please rewrite for better explanation.

[Response 3]: Thanks for the suggestion. We have now rewritten for better explanation (Line 71).

varies 

[Response 4]: The typo has now been corrected.

Please rewrite for better explanation.

[Response 5]: The sentence has now been modified as suggested (Line 101-103).

units please.

[Response 6]: The units has now been added.

Dates please

[Response 7]: Data acquisition dates of Landsat 5 Thematic Mapper (TM) and Landsat 8 Operational Land Imager (OLI) has now been indicated (Line 144-145).

How these classes were extracted. What band combination was used and why?

[Response 8]: Apologise for the confusions caused. The sampled LULC classes were identified based on the field survey and researchers’ previous experience about the study area.  Similarly, the band combination for the classification of Landsat (TM and OLI) images has been mentioned in the revised version of the manuscript (Line 196-199).We have now added the spectral bands combined to create a multi-band composite images of the TM and OLI sensors (196-198).  A single-band image in the visible (blue, green, and red) and near infrared (NIR), and shortwave infrared (SWIR) spectral bands of Landsat 5TM (from Band 1 to Band 5, and Band 7) and OLI (from Band 2 to Band 7) sensors, with a 30 m pixel size, were combined. The spectral bands were chosen for their value in discriminating soil/vegetation, water body, and urban area feature identification.

Is date of Google Earth Images and Landsat Data same?

[Response 9]: Due to a constraint of field data, the Google Earth Image was used to collect references for the 2000 image classification and accuracy assessment. The Google Earth Image and Landsat data is within the same months of dry season, not exactly the same date. We believed that there is no major changes in LULC since the two images were in the same season of 2000. The overall classification accuracy of the Landsat TM classified image of 2000 was higher than the 85% minimum threshold and is satisfactory to conduct LULC change analysis. Therefore, using samples collected from Google Earth for Landsat image classification and accuracy assessment is regarded as a promising approach and attained a good result in the case of our study area.

Areas with a mean annual soil loss rate greater than very low were. Please explain?

[Response 10]: Apologise for the confusion caused. The sentence has now been rewritten as “Areas with a mean annual soil loss rates lower than low medium were rated as tolerable soil loss limit [52]” (Line 255-256).

Please give details of P factor derivations?

[Response 11]: Thanks for the suggestion, we have now added new table (Table 1) to give a clarity for P factor derivations.

Colors of cropland and Shrub land are so close to each other that it is difficult to differentiate between the two. Please use colors to clearly show all the six classes.

[Response 12]: The comment is accepted, we have now enhanced the visibility the maps to clearly illustrate the LULC classes.

It appears that bare land is increasing along the river and decreasing in northern areas of the study area. Why is so? What are the effects of bare land on soil erosion?

[Response 13]: Many thanks for pointing this out.  The increase in bare land along the river in the study area is probably due to the expansion of quarry sites in downstream areas while decreasing in northern parts because of conservation measures put in place over recent years. There are a number of quarry sites at downstream wherein local people extensively extract sand and gravel for construction purpose. This might be the main factors for increases in the bare land cover and soil loss.

Please use actual name of the class instead of alphabets.

[Response 14]: Comment is accepted and correction is made in the current version of the manuscript.

What could be the reason for decrease of water body as it is mainly river?

[Response 15]: During our field visit we have found that there are pond and lakes which has been dried-up in the study area. Since our study used the dry season Landsat satellite images some of the water bodies i.e., ponds and lakes are dried-up and appeared as bare land.

Cropland gain and shrub land loss are similar. Looks like shrub land is being converted into cropland? If so what crop is being cultivated during rainy season and how vegetal cover of cropland is different from shrub land? Please explain.

[Response 16]: Apologise for the confusion caused. The mistake has now been corrected. The shrub land loss accounts for about 27.02% of the total area with about 21.26% of shrubland was changed into cropland.

Reviewer 2 Report

General comments about the manuscript

The manuscript addresses an important question of investigating the impact of land use and cover change on soil erosion. The manuscript reads well, however it can be improved by having it edited for English language. The structure of the manuscript is okay. Methods used are clear. The abstract is well structured and comprehensible. Apart from indicating that model based approached have not been tested before, the contribution to new knowledge by this study is not brought to the fore.

1. Introduction

Full names of acronyms should be written in full first before being used in the text. In the main, acronyms have been given without full descriptions.

2. Materials and Methods

2.2 Data collection should be organized in subsections or paragraphs describing the different datasets used in the study.

2.3 Explanation of how catchment was delineated is missing.

2.3.2 Provide image dates and clarify that single dates imagery were used.

3. Results

Heading should be revised to reflect a discussion is also presented. Discussion of inter-class commission and omissions is lacking.

4. Conclusions

Authors should present what the study set out to investigate before presenting the findings. Conclusions section sounds similar to results. Clarifying the innovation of this study should assist to provide sound conclusions.  

Author Response

Reviewer #2:

General comments:

The manuscript addresses an important question of investigating the impact of land use and cover change on soil erosion. The manuscript reads well, however it can be improved by having it edited for English language. The structure of the manuscript is okay. Methods used are clear. The abstract is well structured and comprehensible. Apart from indicating that model based approached have not been tested before, the contribution to new knowledge by this study is not brought to the fore.

[Response 1]: Many thanks to the Reviewer 2 for the general positive comments and feedback. The manuscript is edited again to improve its grammar. Apologies for the confusion caused, we now have modified the sentence that indicates the model based approached have not been tested before in the background and conclusion sections. Moreover, the theoretical and methodological contributions of our study’s findings have now added. Please see our detailed responses below.

Specific comments:

Introduction: Full names of acronyms should be written in full first before being used in the text. In the main, acronyms have been full given without descriptions.

[Response 2]: We thank the anonymous reviewer for his/her suggestion. The full names of the acronyms has now described first and used consistently throughout the manuscript. 

2 Data collection should be organized in subsections or paragraphs describing the different datasets used in the study.

[Response 3]: We have rearranged sources of spatial datasets used in paragraphs (Line: 165-185).

3 Explanation of how catchment was delineated is missing

[Response 4]: We have improved information on how the study area (catchment) was delineated in the current version of the manuscript (Line: 188-191). We used the raster analysis method based on the terrain data of the digital elevation model (DEM) of 30 meters resolution using the Arc-Hydro extension tools in the ArcGIS environment

3.2 Provide image dates and clarify that single dates imagery were used.

[Response 5]: Apologies for the confusions caused. The satellite image of the Landsat 5 Thematic Mapper (TM) and Landsat 8 Operational Land Imager (OLI) sensors was acquired on14 January 2000 and 20 March 2018, respectively.

Heading should be revised to reflect a discussion is also presented. Discussion of inter-class commission and omissions is lacking.

[Response 6]: Thanks for the suggestion, following the suggestions by another reviewer (Reviewer #1), we have revised the heading of section 3, and now reads as “Results and Discussion.” We have added sentences to elaborate the commission and omission errors among the LULC classes (Line: 396-399).

Authors should present what the study set out to investigate before presenting the findings. Conclusions section sounds similar to results. Clarifying the innovation of this study should assist to provide sound conclusions.

[Response 7]: Many thanks for pointing this out. The conclusions section have been improved for clarity and supported by concrete ideas to illustrate the innovativeness of our study.  

Reviewer 3 Report

This paper is very interesting and it is well analyzed, so I recommend it would be published with minor review.

I recommend that the authors read the few written comments in the text, that I hope increase interest for this paper. The thematic classifications are analyzed correctly and offer the accuracy assessment, but it no clear which sampling method has been used (random points, defined points, by classes, etc.). I also believe that the authors should include more explanations about the causes about land use and land cover evolution and a briefly some possible solutions.

Author Response

Reviewer #3:

General comments:

This paper is very interesting and it is well analysed, so I recommend it would be published with minor review. I recommend that the authors read the few written comments in the text, that I hope increase interest for this paper. The thematic classifications are analysed correctly and offer the accuracy assessment, but it no clear which sampling method has been used (random points, defined points, by classes, etc.). I also believe that the authors should include more explanations about the causes about land use and land cover evolution and a briefly some possible solutions.

[Response 1]: We gratefully thank the reviewer for the thoughtful comments and constructive suggestions, which helped to improve the quality of the manuscript. We have now added the sampling method for collect field data for LULC classification and accuracy assessment (Line 178-180). The attached document ‘peer-review-4692517.v1.pdf’ has been checked and all the comments were incorporated in the revised version of the manuscript.

Specific comments:

and Discussion

[Response 1]: We thank the anonymous reviewer for his/her suggestion, we have revised the heading in section 3 to “Results and Discussion

It is understood, so it is not necessary to repeat it in all percentages

         [Response 1]: Comment is accepted and correction is made in the current version of the manuscript

Before finish, the authors should include more explanations about the causes about land use and land cover evolution, and a briefly some possible solutions. Also in conclusions.

[Response 1]: We included more explanation justifying major causes of LULC evolution and possible solution.  
